# Brain Structure, Cardiorespiratory Fitness, and Executive Control Changes after a 9-Week Exercise Intervention in Young Adults: A Randomized Controlled Trial

**DOI:** 10.3390/life11040292

**Published:** 2021-03-30

**Authors:** Lina Zhu, Qian Yu, Fabian Herold, Boris Cheval, Xiaoxiao Dong, Lei Cui, Xuan Xiong, Aiguo Chen, Hengchan Yin, Zhaowei Kong, Notger Mueller, Arthur F. Kramer, Liye Zou

**Affiliations:** 1School of Physical Education and Sports Science, Beijing Normal University, Beijing 100875, China; zhulina827@mail.bnu.edu.cn (L.Z.); cuilei@bnu.edu.cn (L.C.); 2Exercise & Mental Health Laboratory, Institute of Collaborative Innovation (Sport-Psychology-Education), School of Psychology, Shenzhen University, Shenzhen 518060, China; yuqianmiss@163.com; 3Research Group Neuroprotection, German Center for Neurodegenerative Diseases (DZNE), Leipziger Str. 44, 39120 Magdeburg, Germany; Fabian.herold@dzne.de (F.H.); notger.mueller@dzne.de (N.M.); 4Swiss Center for Affective Sciences, University of Geneva, 1205 Geneva, Switzerland; Boris.Cheval@unige.ch; 5Laboratory for the Study of Emotion Elicitation and Expression (E3Lab), Department of Psychology, FPSE, University of Geneva, 1205 Geneva, Switzerland; 6College of Physical Education, Yangzhou University, Yangzhou 225127, China; DX120190065@yzu.edu.cn (X.D.); DX120190066@yzu.edu.cn (X.X.); 7Faculty of Education, University of Macau, Macao, China; zwkong@um.edu.mo; 8Center for Cognitive and Brain Health, Department of Psychology, Northeastern University, Boston, MA 02115, USA; a.kramer@northeastern.edu; 9Beckman Institute, University of Illinois at Urbana-Champaign, Champaign, IL 61801, USA

**Keywords:** aerobic fitness, grey matter, cortical thickness, executive control, exercise intervention, randomized controlled trial

## Abstract

Cardiorespiratory fitness (CRF) is assumed to exert beneficial effects on brain structure and executive control (EC) performance. However, empirical evidence of exercise-induced cognitive enhancement is not conclusive, and the role of CRF in younger adults is not fully understood. Here, we conducted a study in which healthy young adults took part in a moderate aerobic exercise intervention program for 9 weeks (exercise group; n = 48), or control condition of non-aerobic exercise intervention (waitlist control group; n = 72). Before and after the intervention period maximal oxygen uptake (VO_2max_) as an indicator of CRF, the Flanker task as a measure of EC performance and grey matter volume (GMV), as well as cortical thickness via structural magnetic resonance imaging (MRI), were assessed. Compared to the control group, the CRF (heart rate, *p* < 0.001; VO_2max_, *p* < 0.001) and EC performance (congruent and incongruent reaction time, *p* = 0.011, *p* < 0.001) of the exercise group were significantly improved after the 9-week aerobic exercise intervention. Furthermore, GMV changes in the left medial frontal gyrus increased in the exercise group, whereas they were significantly reduced in the control group. Likewise, analysis of cortical morphology revealed that the left lateral occipital cortex (LOC.L) and the left precuneus (PCUN.L) thickness were considerably increased in the exercise group, which was not observed in the control group. The exploration analysis confirmed that CRF improvements are linked to EC improvement and frontal grey matter changes. In summary, our results support the idea that regular endurance exercises are an important determinant for brain health and cognitive performance even in a cohort of younger adults.

## 1. Introduction

Brain and cognition health among adolescents and young adults can influence academic achievement and overall health throughout the life span [1,2]. Thus, exploring predictors and modifiers of brain health in the young age is crucial. In this context, there is a growing interest to understand the influence of lifestyle factors such as regular physical exercise promoting a relatively high level of cardiorespiratory fitness (CRF) on health (e.g., brain health, [3]). CRF is typically operationalized by maximal oxygen consumption (VO_2max_) and is related to the capabilities of different physiological systems such as the respiratory systems [4]. Remarkably, low levels of CRF in early adulthood are associated with higher risks of cardiovascular disease in late life, and it also negatively affects psychological functioning (i.e., depression, anxiety) and cognitive functioning becoming evident in accelerated cognitive decline and brain atrophy in later years [5,6,7,8,9].

Indeed, there is mounting evidence in the literature indicating that higher CRF levels and regular physical exercise are associated with better cognitive performance, especially in children and older adults [10,11]. For example, executive control, a cognitive domain associated with “top-down” regulation of goal-directed behavior [12,13], is positively associated with physical exercising and CRF level in both children and older adults [14,15,16,17,18,19,20,21,22]. However, the effects of exercise interventions in early adulthood on EC performance are not fully understood [23,24]. This gap in the literature could be related to the view that young adults perform, in general, quite well on cognitive tasks, leaving limited space for physical exercise-related improvements in cognitive performance. However, as it remains unclear whether the evidence obtained in children and older adults can be generalized to younger adults, further investigations have been recommended in this age group [24].

In the literature, it has been reported that physical exercise interventions (e.g., endurance training leading to an increase in CRF) are increasingly recognized as a promotive factor for grey matter development in children and adolescents, and as a protective factor against gray matter atrophy in the elderly [25,26,27,28,29,30]. Indeed, evidence demonstrates a positive association between CRF and grey matter volume (GMV) especially in elderly subjects [25,26,27,28,29,30]. Higher levels of CRF are associated with increased hippocampal and prefrontal cortex volume as well as better cognitive performance (i.e., memory, EC) in older individuals [30,31]. At the other end of the age spectrum (i.e., children), CRF and cortical volume and morphology (i.e., hippocampus, basal ganglia, frontal cortex), as well as cognitive functioning (i.e., working memory, EC), exhibit a positive correlation [31,32,33]. Despite gradually mature brain maturation in young adulthood, several studies have found a positive association between CRF and grey matter plasticity (e.g., entorhinal volume, hippocampus) [34,35,36]. However, the vast majority of the studies investigating the relationship between CRF, brain structure, and cognitive performance are cross-sectional studies that do not allow causal conclusions. Furthermore, the majority of the available studies focused on children or older adults, and thus, our knowledge concerning younger adults is meager [37]. Therefore, longitudinal studies investigating the relationship between changes in CRF level, brain structure, and cognitive performance in response to endurance training are necessary to better understand exercise–cognition interaction but are currently relatively sparse in younger adults [24].

Based on the cardiorespiratory fitness hypothesis, which stipulates that an increase in CRF level improves cognitive performance and triggers positive brain changes [38,39], we hypothesize that endurance training aiming to improve CRF level also induces positive effects on cognitive performance and the brain even in a cohort of younger adults. Thus, this study aims to investigate whether a nine-week endurance exercise program changes CRF level, brain structure, and cognitive performance. To quantify exercise-related changes in brain structure, we use Voxel-based morphometry (VBM) permitting a general quantification of GMV change, and surface-based morphometry (SBM) quantifying thickness changes in the distance between the gray/white matter boundary and the cortical surface [40,41]. Based on the positive neurobehavioral relationships between CRF level, brain structure, and cognitive performance being reported in previous studies [29,31], the present study also explores whether changes in CRF, if present, are linked to changes in GMV, cortical thickness, and EC performance.

## 2. Methods

### 2.1. Study Design

This randomized control trial was based on a two × two mixed factorial design and used group (exercise vs. controls) as a between-subject factor and time (post-exercise vs. pre-exercise) as a within-subject factor (repeated measures). The experiment was conducted at Yangzhou in China, with ethics approval from the Ethics and Human Protection Committee of the Affiliated Hospital of Yangzhou University (2017-YKL045-01). All study procedures were in accordance with the latest version of the Declaration of Helsinki.

### 2.2. Procedures

The participants were randomly allocated to the exercise and control group (see Figure 1). As further detailed in Section 2.4, the exercise group performed four exercise sessions in a week, whereas the control group was placed on a waitlist and was advised to not participate in any endurance, stretching, and toning training during the intervention period. All participants performed cognitive testing (i.e., Flanker test) and structural neuroimaging before (Time 1) and after nine weeks of intervention (Time 2). Changes in anthropometry (i.e., body mass index [BMI]) and the cardiorespiratory fitness level were also evaluated before and after the intervention period. An overview of the study procedures is provided in Figure 1.

### 2.3. Participants

We conducted a statistical power analysis to determine the required sample size for a repeated-measures ANOVA including group (exercise vs. controls) as a between-subject factor and time (post-exercise vs. pre-exercise) as a within-subject factor a priori. In this context, 100 participants (50 per group) are needed to achieve a power of 0.93 for the within-subjects main effect at an effect size of 0.25, and a power of 0.93 for the interaction effect at an effect size of 0.25. In total, one hundred and twenty-three college students were randomly assigned to the exercise group or the control group in the current study. The results reported here contain data from 120 young adults, as data from three participants were discarded due to insufficient data quality. The exercise group’s final sample included 48 participants, while the control group contained 72 participants. The following inclusion criteria were used: (1) 18–20 years of age; (2) right-handedness, and (3) normal vision without color blindness. On the other hand, the exclusion criteria included: (1) any mental and/or physical disorders that limited physical activity and/or the research results; (2) substance abuse (i.e., drug, nicotine, alcohol); (3) MRI contraindications (i.e., metallic implants, claustrophobia, pacemakers, or contrast allergy). All participants received a compensation for their participation in the study. An overview of the assessed variables is displayed in Table 1.

### 2.4. Fitness Intervention

The exercise group was enrolled in the school-based physical exercise courses, while the control group was placed on a waitlist and continued their normal school activities [42]. The exercise and training variables used in this exercise intervention followed the recommendations outlined in the American Exercise Medicine Association guidelines for young adults [43]. In this regard, the exercise group conducted four exercise sessions a week for nine weeks. In particular, all exercise classes started at 6.30pm and were conducted on the following days: a 60 min fitness exercise sessions on Monday, Tuesday, and Thursday and a 45 min aerobic running session on Sunday. The lesson started with a 10 min warming-up period and ended with a 10 min cooling-down period, including balance, flexibility, and relaxation exercises. The intervention includes competitive and non-competitive sports and games to increase social interactions, which may enhance cognitive stimulation and movement skill proficiency [42]. To track the exercise intensity, heart rate (HR) was monitored continuously using a heart rate monitor system worn by randomly selected 6 participants in each lesson (BHT GOFIT, a team training system developed based on Polar and tracks the real-time heart rate of each student to allow an accurate exercise intensity monitoring (<Introduction for BHT GOFIT> http://www.bhttech.com (accessed on 10 March 2021)). The exercise intensity was progressively increased by increasing the difficulty level of the exercises to ensure a moderate exercise intensity (64–77% maximum heart rate) across the intervention period. The control group was set on a waitlist and was advised to not engage in any endurance, stretching, and toning training during the intervention period.

### 2.5. Cardiorespiratory Fitness Testing

Before the intervention period and after the intervention, both groups performed a graded exercise test (GXT) to determine the cardiorespiratory fitness level operationalized by the participant’s maximal oxygen consumption (VO_2 max)_. VO_2 max_ values correspond to one’s maximum ability to consume and utilize oxygen during physical exertion and higher values indicating better cardiorespiratory fitness. The VO_2 max_ was assessed via a computerized indirect calorimetry system during a modified cycle ergometer test (American College of Sports Medicine, [43]). At the beginning of the GXT, the participants performed a warm-up period of two minutes cycling at low intensity. After this warm-up period, the participants cycled at a constant speed, and the initial load was 50W, the cycling rhythm was 55–60 r/min, and then increased by 50w every three minutes until exhaustion. Averages for oxygen uptake (VO_2_) were assessed every 30 s. A measure of perceived exertion was attained every two minutes using the participants’ RPE scale of perceived exertion. Maximum oxygen consumption (VO_2 max_) was measured in milliliters per kilogram per minute (mL/kg/min) and based on a maximal effort, which was defined by accomplishing, at least, two of the following four criteria: (1) age-defined maximum heart rate norms (i.e., heart rate > 85% of predicted maximum heart rate)), (2) a respiratory exchange ratio (the ratio between carbon dioxide and oxygen percentage) greater than 1.1, (3) a rating greater than eight on the RPE scale of perceived exertion, or (4) a plateau in VO_2_ despite an increase in workload [34].

### 2.6. Flanker Task

A modified Eriksen Flanker task was employed to examine the executive control. The Eriksen Flanker task is sensitive to the effects of acute exercise and chronic exercise [44,45]. The Flanker task involved two types of trials: congruent and incongruent. The congruent trials consisted of a horizontally arranged array of the same five letters (e.g., LLLLL or FFFFF); the incongruent trials consisted of a horizontally arranged array of five letters in which the middle letter was different (e.g., LLFLL or FFLFF). During the Flanker task, the participant was instructed to focus on the fixation cross and, as soon as the stimulus appears, press the correct button (“F” or “L”) with their left or right index finger, respectively, based on the middle letter presented in the trial. A fixation cross (+) was first presented for 500 ms at the center of the screen to attract the participant’s attention. Then, either a congruent or an incongruent letter set was displayed for 1000 ms. The stimulus onset asynchrony was set at 2000 ms. All stimuli were presented on a white background. The participants were instructed to react to each trial as quickly and accurately as possible by responding to the center letter in each array. Pressing the wrong button and responding within 150 ms or timeout were each considered incorrect responses. Participants were asked to perform 12 practice trials and then complete two blocks of 48 trials each, with a 1 min rest interval between the blocks. The congruent and incongruent trials were presented in random order with equal probability in each block. The total task duration was approximately 6 min. The response times in the congruent and incongruent trials were recorded and used to evaluate the inhibition performance. Shorter response time (RT) and higher accuracy reflected better cognitive performance.

### 2.7. MRI Acquisition and Structural Image Analysis

MRI data were acquired using the 3 Tesla Discovery MR 750 MRI scanner (GE Healthcare; United States), which is situated in the Hospital of Yangzhou University. Structural data were acquired with a high-resolution magnetization prepared rapid acquisition gradient-echo T1-weighted sequence (echo time (TE) = 3 ms; slice thickness = 1 mm; field of view = 256 mm; matrix = 64 × 64; voxel size: 1.0 × 1.0 × 1.0 mm; flip angle = 12°).

#### 2.7.1. Voxel-Based Morphometry

T1 images were segmented in grey matter and white matter map using the CAT12 toolbox (<Introduction for CAT12 Toolbox> http://www.neuro.uni-jena.de/cat/(accessed on 10 March 2021)) that is based on SPM (<Introduction for SPM Software> https://www.fil.ion.ucl.ac.uk/spm/(accessed on 10 March 2021)). Briefly, we spatially registered the 3DT1 images to the tissue probability maps (TPM) and segmented them into gray matter (GM), white matter (WM), and cerebrospinal fluid. We then performed the affine registration to the stereotactic Montreal Neurological Institute (MNI) space using the ICBM152 space [46]. To improve registration accuracy, a group-specific template was established using the lie algebra diffeomorphic anatomical registration through exponentiated (DARTEL) algorithm in statistical parameter mapping, and a non-linear transformation of individuals to the template was calculated [47]. Finally, the grey matter volume images were smoothed with a Gaussian kernel of 8mm FWHM.

#### 2.7.2. Cortical Thickness Estimation

Surface-based neuroimaging analysis techniques were used to quantify cortical thickness, which estimated the cortical thickness at each point across the cortical mantle by calculating the distance between the gray/white matter boundary and the cortical surface. CAT12 is a fully automatic method that measures cortical thickness and reconstructs the central surface in one step. The program uses tissue segmentation to estimate the WM distance and uses the neighbor relationship described by the WM distance to project a local maximum (equal to the thickness of the cortex) to other GM voxels. The projection-based thickness (PBT) allows the management of partial volume information, sulcal blurring, and sulcal asymmetries without the need for explicit groove reconstruction through bone or thinning methods. The merged surface data for the right and left hemispheres were then resampled and spatially smoothed using the default 15 mm full-width-at-half-maximum (FWHM) Gaussian smoothing kernel. Based on the quality of their cortical surface reconstructions, we excluded the data of one individual in the exercise group and of two individuals in the control group (see Figure 1). For each subject, the average of each cluster entailing significant differences in the GM and cortical thickness were then extracted and used for further statistical analyses.

### 2.8. Statistical Analysis

All statistical analyses were performed using the Statistical Package for the Social Sciences (SPSS; SPSS Inc., Chicago, IL, USA) version 25.0 for Windows. The statistical significance threshold was set at *p* < 0.05. We compared the post-exercise VO_2max_ between the two groups with ANCOVA analyses, controlling for baseline VO_2max_. A repeated-measures analysis of variance (ANOVA) was performed to evaluate the intervention effect on RT and accuracy. Bonferroni correction was applied to correct for multiple comparisons. Using a two × two repeated measures ANOVA (flexible factorial design in SPM) with a between-subjects factor group (exercise vs. controls) and a within-subjects factor time (pre-exercise vs. post-exercise), we investigated which cortical regions exhibited changes in grey matter (e.g., increasing or decreasing volume and thickness) in response to the intervention while correcting for age, sex, baseline VO_2max_, and total intracranial volume (TIV). All fMRI results were not corrected by false discovery rate (FDR)and family-wise error (FWE).

In order to investigate possible neurobehavioral relationships, we performed correlation analyses such as partial correlations (controlling for age and sex). Thereto, we computed change scores (Time 2 minus Time 1) for cognitive performance (i.e., RT) and neural correlates (GM changes and cortical thickness changes). We rated the correlation coefficients as follows: 0 to 0.19: no correlation; 0.2 to 0.39: low correlation, 0.40 to 0.59: moderate correlation; 0.60 to 0.79: moderately high correlation; ≥ 0.80: high correlation [48].

In addition, PROCESS 3.3, which is freely available computational macros for SPSS integrating the mediation and moderation analysis, was used to perform a mediation analysis. We used Model 4 of the PROCESS template to test the mediating role of brain structure change between VO_2max_ and incongruent RT. The macros calculate total effects, direct effects, and indirect effects with bootstrap confidence interval based on 5,000 resamples. In accordance with the literature, we concluded that the indirect effect was statistically significant if the confidence interval did not contain zero [49,50].

## 3. Results

### 3.1. Demographic Analyses

No statistically significant differences were observed between the groups in terms of sex (χ^2^ = 1.250, *p* = 0.264), BMI (t (118) = 0.815, *p* = 0.417, Cohen’s d = −0.152), age (t (118) = 1.393, *p* = 0.166), congruent RT (t (118) = −1.332, *p* = 0.185, Cohen’s d = 0.248), congruent accuracy (t (47) = −0.269, *p* = 0.788, Cohen’s d = 0.050), and incongruent accuracy (t (118) = −0.095, *p* = 0.924, Cohen’s d = 0.018) at the baseline (Table 1 and Table 2). As differences were shown in between-group VO_2max_ (t (118) = −2.959, *p* = 0.004, Cohen’s d = 0.551) and incongruent RT (t (118) = 2.881, *p =* 0.002, Cohen’s d = −0.537), both factors were regarded as covariates in the later analyses of intervention effects.

### 3.2. Cardiorespiratory Fitness

A significant increase in HR (F (2, 105) = 280.592, *p* < 0.001, partial *η*^2^ = 0.842) was observed during the period of fit-intervention (136.641 ± 7.773 bpm), compared to the periods of warming up (82.856 ± 10.74 bpm) and cooling-down (111.067 ± 10.137 bpm). During the 9-week intervention, the average VO_2max_ scores increased from 21.566 to 28.066 in the exercise group, whereas the reverse applied in the control group (pre-intervention: 24.616; post-intervention: 24.347); a significant group effect was revealed by ANCOVA analysis (F (1,118) = 115.564, *p* < 0.001), partial *η*^2^ = 0.497.

### 3.3. Executive Control Function

For the congruent accuracy, the mixed-design ANOVA revealed no main effects of group and time; no significant interaction of group by time was found (F (1, 118) = 0.021, *p* = 0.886, partial *η*^2^ = 0.0001). Similar results were obtained with the incongruent accuracy (F (1, 118) = 0.906, *p* = 0.343, partial *η*^2^ = 0.008).

For the congruent RT, the mixed-design ANOVA revealed no main effects of group (F (1, 118) = 47.689, *p*< 0.001, partial *η*^2^ = 0.288) and interaction of group by time (F (1, 118) = 47.689, *p* < 0.001, partial *η*^2^ = 0.288), whereas a main effect of time was observed (F (1, 118) = 47.689, *p* < 0.001, partial *η*^2^ = 0.288). For the incongruent RT, the significant group effect (F (1,118) = 16.672, *p_Bonferroni_* < 0.001, partial *η*^2^ = 0.125) indicated faster incongruent RT induced by the exercise intervention.

### 3.4. Brain Structural Alteration

#### 3.4.1. Grey Matter Analysis

Repeated measures ANOVA analysis revealed the between-group difference on GMV (group × time interaction, *p* < 0.05, FDR-corrected) of left medial superior frontal gyrus (medial FG.L) (Figure 2). While a slight increase in the medial FG.L GMV was shown in the exercise group (F (1, 2) = −1.443, *p* > 0.05), the control group proved the contrary (F (1, 2) = 2.728, *p* < 0.01).

#### 3.4.2. Cortex Thickness Analysis

The repeated-measures ANOVA analysis revealed a significant group × time effect (*p*_vertex_ < 0.001 uncorrected, *p_cluster_* < 0.05, FWE-corrected) in the left lateral occipital cortex (LOC) and precuneus (PCUN) (Table 3; Figure 3). LOC.L (F (1, 2) = −4.831, *p <* 0.001) and PCUN.L (F (1, 2) = −2.535, *p <* 0.05) thickness increased in the exercise group following the exercise intervention, whereas significant alterations were not found in the control group.

### 3.5. Correlation Analysis

The region of interest approach was used to correlate exercise-related structure changes in GMV and thickness with incongruent RT change, controlling for age and sex. Correlations between VO_2max_, LOC.L thickness, and incongruent RT (*p* < 0.01) were observed in the exercise group (Figure 4) but not in the control group (*p* > 0.05). Additionally, the changes in GMV of medial SFG.L were positively associated with changes in VO_2max_ (Figure 4). Furthermore, GMV changes in the medial SFG were found to mediate the effects of improved AF on executive control performance (i.e., incongruent RT) (Figure 5).

## 4. Discussion

The present study investigated the intervention effects of a nine-week endurance exercise program on young adults’ CRF level, brain structure (i.e., GMV and cortical thickness), and a measure of executive control performance. Increases in CRF level and GMV in left medial FG were observed in response to a 9-week endurance intervention, whereas the inverse trend was noticed in the control group. Likewise, the cortical thickness in LOC.L and PCUN.L significantly increased in the exercise group but did not in the control group. Additionally, the mediation analysis indicated that frontal gyrus GMV increment mediated the effects of AF on executive control performance.

Our observation of an increase in EC performance after nine weeks of endurance training is in line with findings of previous studies noticing an improvement of executive functioning in younger adults after six months of endurance training [51]. Moreover, our findings support the cardiorespiratory fitness hypothesis predicting that a higher level of cardiovascular fitness benefits cognitive performance [38,39].

According to Stillman et al., regular physical activity such as endurance training triggers changes on multiple levels (e.g., Level 1—cellular and molecular changes; Level 2—structural and functional brain changes; Level 3—socioemotional changes), which, in turn, drive the changes in cognitive performance [52].

In line with this theoretical assumption, our study revealed that the GMV in the left medial FG increased in the endurance training group. This finding broadens our knowledge regarding the effects of endurance training on brain structure as previous studies investigating the effect of long-term endurance training in younger adults focused on the hippocampal region [53,54]. These studies observed that long-term endurance training increases the volume of a subfields of the hippocampus (i.e., CA4-DG, dentate gyrus/CA3) [53,54]. Notably, besides the connections with motor functions, the medial FG was also reported to be a brain region associated with high-level executive function and decision-related process, with lateralization in response executing and imaging [55,56]. To be specific, the medial FG was biased toward the left hemisphere during the “what” and “when” task and toward the right hemisphere during the “where” task [56].

We also observed an increase in cortical thickness in LOC.L and PCUN.L in the exercise group, but not in the control group. The LOC., a high position in the hierarchy of visual areas, plays an important role in visual information maintaining and object recognition (domains of visual working memory [57]). The PCUN. is reported to modulate cognitive functions by interconnecting parietal and prefrontal regions “small-world network” hub [58]). In the present study, the increased cortical thickness in LOC.L and PCUN.L is not fully in line with a previous study indicating that long-term endurance training in younger adults resulted in increased cortical thickness in frontal brain regions [51]. Such inconsistency in the observations could be related to the training duration. In the study of Stern et al. [51], the participants conducted an endurance intervention lasting six months, while the participants in our study trained for nine weeks. However, this explanation remains speculative, as there is currently no study available that investigated a possible dose-response relationship between training duration and changes in cortical thickness in specific brain regions in younger adults [24].

Our finding that the increase in the GMV of the left medial FG mediates the positive relationship between the CRF level and EC in the exercise intervention groups fits with the idea of Stillman et al. [24], assuming that the effects of regular physical activity (e.g., endurance training) are mediated by changes on different neurobiological levels (e.g., Level 2—structural and functional brain changes). Moreover, this finding corresponds with the observation of previous studies demonstrating that in older adults, changes in brain structure (e.g., hippocampus [29] or prefrontal cortex [31]) mediates the relationship between the exercise-induced increase in CRF level and cognitive performance. Thus, our finding complements the existing literature by showing that even in younger adult’s exercise-induced changes in brain structure are an important mediator between physical fitness (e.g., CRF) and cognitive performance. Additionally, as the psychological trauma usually has negative impacts on the immunological phenotype of microglia of the frontal cortex [55], the unexpected decrease in grey matter volume in the control group may be induced by participants’ stress reaction during the final examination [14,59,60].

## 5. Limitation

There are some limitations in this study that need to be acknowledged and further discussed. Firstly, the baseline differences in VO_2max_ and congruent RT between the exercise and control groups may reduce the reliability of intervention effects, although we have considered both variables as covariates in our statistical analysis. Secondly, we did not control for psychological confounders (i.e., stress induced by final exam) such as expectation effects, which might have influenced our results. Such important psychological variables should be considered in future investigations [61,62]. Thirdly, based on the fact that we have used a variety of group exercises to create a motivating and ecologically valid setting for our endurance training intervention, an exact a priori prescription of exercise variables and post hoc determination of the dose is difficult (e.g., exercise intensity and intermittency) [63,64]. Fourthly, the exercise group and control group were not matched for sex, which was a bias of the present study. Thus, similar large-scale studies with sex-matched groups are needed. Fifthly, due to the time limitation, only six participants were randomly selected for heart rate measuring, which brought hidden security risks. To at least partially take this circumstance into account, we tracked heart rate changes via a heart rate monitoring system continuously throughout the exercise sessions.

## 6. Conclusion

In summary, the young adults in the endurance exercise group demonstrated a significant increase (i) in GMV in the left medial frontal gyrus, (ii) thickness in the left lateral occipital cortex, and (iii) thickness in the left precuneus after the exercise intervention. Notably, the change in GMV in the left medial frontal gyrus in response to the exercise intervention mediates the relationship between the changes in CRF and cognitive performance improvements. These findings suggest that even in younger adults, endurance exercise can trigger neurobiological processes leading to structural brain changes such as increased volume in prefrontal regions, which, in turn, drive cognitive performance improvements. Thus, the present study supports the cardiorespiratory fitness hypothesis, as they confirm the important role of regular endurance exercise to promote brain health and cognitive performance in younger adults.

## Figures and Tables

**Figure 1 life-11-00292-f001:**
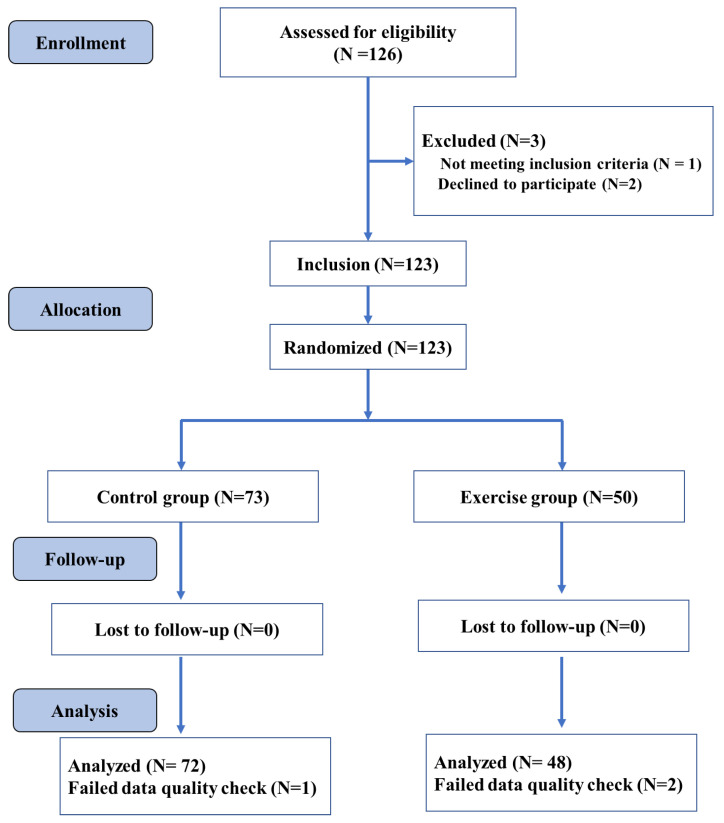
Overall study design flowchart.

**Figure 2 life-11-00292-f002:**
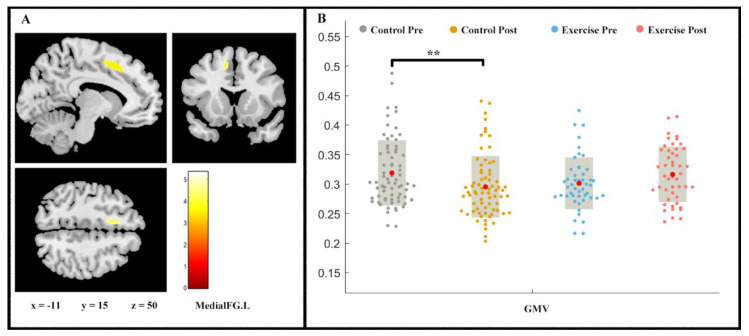
Volume changes in grey matter during 9 weeks of exercise intervention. Note. (**A**) Significant (*p* < 0.05, FDR-corrected) longitudinal grey matter volume increases; (**B**) grey matter volume (GMV) in medial FG.L for two groups at pre- and post-exercise. x, y, z = coordinates in MNI space. ** means *p* < 0.01.

**Figure 3 life-11-00292-f003:**
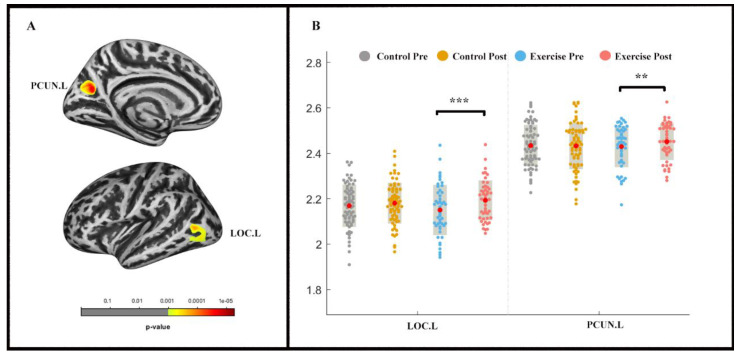
Cortical thickness changes during 9 weeks of exercise intervention. Note. (**A**) Significant (*p*_vertex_ < 0.001 uncorrected, *p_cluster_* < 0.05, FWE-corrected) longitudinal cortical thickness increases; (**B**) cortical thickness in left lateral occipital cortex (LOC.L) and the left precuneus (PCUN.L) for two groups at pre-and post-exercise. ** means *p* < 0.01,*** means *p* < 0.001.

**Figure 4 life-11-00292-f004:**
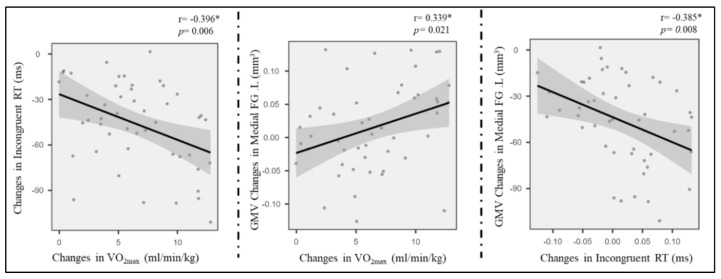
Pairwise correlations between AF change, grey cortical changes, and incongruent RT enhancement. * means *p* < 0.05.

**Figure 5 life-11-00292-f005:**
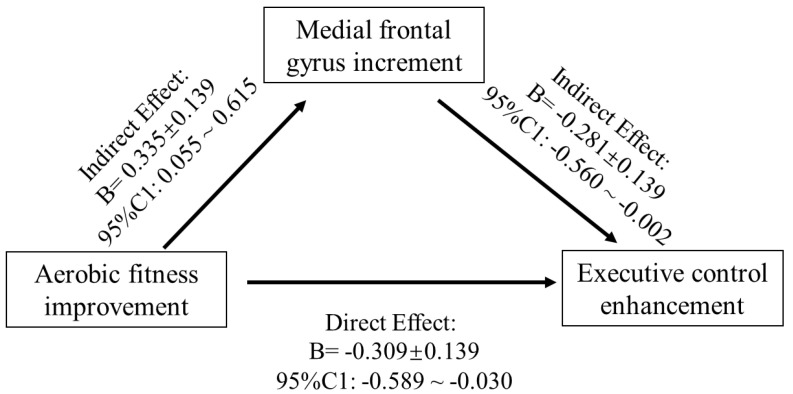
Mediation model of the frontal gyrus GMV increase between AF change and EC enhancement.

**Table 1 life-11-00292-t001:** Characteristics of study participants.

	Control Group	Exercise Group	T Value	*p* Value	Effect Size (Cohen’s d)
Sex (female)	72(33)	48(27)	1.250	0.264	-
Age (years)	18.500 ± 0.692	18.670 ± 0.559	1.393	0.166	–0.260
BMI (m/kg^2^)	21.305 ± 2.812	21.772 ± 3.433	0.815	0.417	–0.152
eTIV (mm^3^)	1496.664 ± 135.511	1467.893 ± 128.707	1.162	0.247	0.217

Note: Values indicate mean ± standard deviation (SD). eTIV, estimated total intracranial volume.

**Table 2 life-11-00292-t002:** Cardiorespiratory Fitness and Inhibitory Control at Baseline and Post-Test.

	Control Group	Exercise Group	T Value	*p* Value	Effect Size (Cohen’s d)
*Pre−intervention*					
VO_2max_(ml/kg/min)	24.616 ± 5.268	21.566 ± 5.907	−2.959	0.004 **	0.551
Congruent RT (ms)	484.419 ± 47.740	473.195 ± 41.082	−1.322	0.185	0.248
Congruent Accuracy (%)	0.956 ± 0.046	0.953 ± 0.053	−0.269	0.788	0.050
Incongruent RT (ms)	516.756 ± 49.977	539.800 ± 29.179	2.881	0.005 **	−0.537
Incongruent Accuracy (%)	0.943 ± 0.052	0.942 ± 0.051	−0.095	0.924	0.018
*Post−intervention*					
VO_2max_(ml/kg/min)	24.347 ± 5.043	28.066 ± 5.587	3.790	0.000 ***	−0.706
Congruent RT (ms)	457.828 + 47.675	441.097 ± 42.309	−1.968	0.051	0.367
Congruent Accuracy (%)	0.961 ± 0.045	0.949 ± 0.049	−1.465	0.145	0.273
Incongruent RT (ms)	513.292 ± 57.114	493.679 ± 30.675	−2.177	0.031 *	0.406
Incongruent Accuracy (%)	0.946 ± 0.043	0.946 ± 0.045	0.069	0.945	−0.013

Note: Values indicate mean ± standard deviation (SD). RT, reaction time. * means *p* < 0.05, ** means *p* < 0.01,*** means *p* < 0.001.

**Table 3 life-11-00292-t003:** Cortical thickness: results of the repeated-measures ANOVA analysis.

*p*-Value	Cluster Size	Overlap of Atlas Region	
0.00014	962	68%	Lateral occipital cortex
0.00002	839	75%	Precuneus
		25%	Cuneus

*p*_vertex_ < 0.001 uncorrected, *p_cluster_* < 0.05, FWE-corrected.

## Data Availability

The data presented in this study are available on request from the corresponding author. The data are not publicly available due to privacy of participants.

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
