# Peer review of "Brain Structure, Cardiorespiratory Fitness, and Executive Control Changes after a 9-Week Exercise Intervention in Young Adults: A Randomized Controlled Trial"

_life, 2021, doi:10.3390/life11040292_

Round 1
Reviewer 1 Report
Summary:
The authors conducted a randomized control study involving healthy young adults: an exercise group (n48, which took place to a 9-week exercise program) and a control group (n72, non-aerobic exercise 9 intervention). They assessed maximal oxygen uptake, Flanker task, grey matter volume and cortical thickness before and after the intervention period. They found increase in gray matter volume in left medial frontal gyrus in the exercise group, whereas reduction in the control group and cortical thickness increase in left lateral occipital cortex and left precuneus in the exercise group. Lastly they found correlation between Cardiorespiratory fitness, executive control and frontal grey matter changes supporting the idea that regular exercises important for brain health and cognitive performance.
The article is well written, an overall good read, however some concerns need to be aknowledged:
Abstract: there is an error on subjects number, maybe the groups were inverted. Please correct or clarify.
Introduction:ok
Methods: The groups were not matched for sex. The author have to clarify why, since this kind of study could be influenced by sex and not gender.
In Table 1: since the table describes groups characteristics, the post-treatments data should be described in the result section with corresponding p values ; RT, what does you mean for reaction time? please define it in the text since it has been not mentioned before and after the table, but only at line 190 as response time.
MRI: line 193. please specify from which vendor the scanner was manufactured by;
Results: although the statistical power of correlation analyses were not so good as the correlation graphics, the explanation makes sense.
Discussion: need to be improved explaining better the results; in particular it would be interesting to better explain the role of left medial frontal gyrus in executive control, and to make an hypothesis for higher LOC.L and PCUN.L thickness in exercise group.
Author Response
The article is well written, and an overall good read, however, some concerns need to be acknowledged:
Abstract: there is an error on subjects number, maybe the groups were inverted. Please correct or clarify.
Answer: Thanks for your reminder and we have corrected the subject number in the abstract.
Introduction: ok
Methods: The groups were not matched for sex. The author has to clarify why, since this kind of study could be influenced by sex and not gender.
Answer: Thanks for your kind suggestion. We are sorry to misuse the word “gender” and the word “sex” has been used to take the place of “gender”. In the design of this study, we did consider the influences brought by biological character (male/ female) of human beings.
In Table 1: since the table describes group characteristics, the post-treatments data should be described in the result section with corresponding p values; RT, what do you mean for reaction time? please define it in the text since it has been not mentioned before and after the table, but only at line 190 as response time.
Answer: Thanks for your kind suggestion. The corresponding p values, t values, and effect size (Cohen’s d) have been inserted in Table 1. Regarding the RT, we have defined it in the note of Table 1.
MRI: line 193. please specify from which vendor the scanner was manufactured;
Answer: Thanks for your kind suggestion and the information related to the scanner as followed has been provided.
“ MRI data were acquired using the 3 Tesla Discovery MR 750 MRI scanner (GE Healthcare; United States), which is situated in the Hospital of Yangzhou University.”
Results: although the statistical power of correlation analyses was not so good as the correlation graphics, the explanation makes sense.
Discussion: need to be improved explaining better the results; in particular it would be interesting to better explain the role of the left medial frontal gyrus in executive control and to make a hypothesis for higher LOC.L and PCUN.L thickness in the exercise group.
Answer: Thanks for your suggestion and the discussion has been revised as followed to better explain the results.
[1] Explaining the role of the left medial frontal gyrus in executive control
“ Notably, besides the connections with motor functions, the medial FG was also reported to be a brain region associated with high-level executive function and decision-related process, with lateralization in response executing and imaging. To be specific, the me-dial FG was biased toward the left hemisphere during the ‘what’ and ‘when’ task and toward the right hemisphere during the ‘where’ task.”
[2] Hypothesis for higher LOC.L and PCUN.L thickness in the exercise group.
“We also observed an increase of cortical thickness in LOC.L and PCUN.L in the exercise group, but not in the control group. The LOC., high position in the hierarchy of visual areas, plays an important role in visual information maintaining and object recognition (domains of visual working memory). And the PCUN. is reported to modulate cognitive functions by interconnecting parietal and prefrontal regions ‘small-world network’ hub).”

Reviewer 2 Report
The authors presented a interesting study about the effects of an endurance exercise program on your adults an the influence to CRF level, brain structure and measures of executive control performance.
Main comments:
- Please shorten the introduction and discuss your results in more detail in the discussion section
- Please insert a Limitations section and insert the information given from line 372 onwards. The same applies to the manuscript text from point 384 onwards. Please exclude this part as a conclusion section
- Fitness intervention: You explain that you have randomly selected 6 participants in each lesson for hear rate measuring. Were the same participants in all lessons or did they switch between lessons? Were you not all of them? This approach does not appear to be valid with a total of 48 participants in the intervention group. This appears to be a bias in the study?
Minor comments:
- line 32: please put the reference in brackets.
- line 55: Please underline your statement by a reference
- line 68: " but see 37"--> please change the wording
- Table 1: Please include the p-values of the variables
- line 133: Please include a reference to your statement
Author Response
The authors presented a interesting study about the effects of an endurance exercise program on your adults an the influence to CRF level, brain structure and measures of executive control performance.
Main comments:
- Please shorten the introduction and discuss your results in more detail in the discussion section
Answer: Thanks for your kind suggestion. The introduction has been shortening and more details have been provided in the discussion section.
- Please insert a Limitations section and insert the information given from line 372 onwards. The same applies to the manuscript text from point 384 onwards. Please exclude this part as a conclusion section.
Answer: Thanks for your suggestion and it has been revised.
- Fitness intervention: You explain that you have randomly selected 6 participants in each lesson for heart rate measuring. Were the same participants in all lessons or did they switch between lessons? Were you not all of them? This approach does not appear to be valid with a total of 48 participants in the intervention group. Does this appear to be a bias in the study?
Answer: Thanks for your kind suggestion. The six randomly selected participants switched between lessons. For a sample with 48 participants, this approach may be not valid and we have admitted it in the limitation section.
Minor comments:
- line 32: please put the reference in brackets.
Answer: Thanks for your remind and the brackets have been added.
- line 55: Please underline your statement by a reference
Answer: Thanks for your suggestion and the references have been added.
- line 68: " but see 37"--> please change the wording
Answer: Thanks for your suggestion and it has been revised.
- Table 1: Please include the p-values of the variables
Answer: Thanks for your suggestion and it has been revised.
- line 133: Please include a reference to your statement
Answer: Thanks for your suggestion and the references have been added.
Congratulation! Overall this is a successful study with interesting and important results.

Round 2
Reviewer 1 Report
The article has been improved, however some concerns need to noticed:
-authors correctly changed "gender" with "sex", but the exercise group and control group were matched for sex? in the exercise group there were more female compared with control group, and this could be a bias. please add in limitation section
-please place the pre-intervention and post-intervention section of the table 1 in the results section.
-discussion: please correct this sentence "In the previous study, the increased cortical thickness in LOC.L and PCUN.L is not fully in line with a previous study investigating the effect of long-term endurance training in younger adults as they noticed an increase in cortical thickness in frontal brain regions [51]".
Author Response
[1] The authors correctly changed "gender" with "sex", but the exercise group and control group were matched for sex? in the exercise group there were more females compared with the control group, and this could be a bias. please add in the limitation section
Answer: Thanks for your reminder and we have added this as a limitation.
“Fourthly, the exercise group and control group were not matched for sex, which was a bias of the present study.”
[2] please place the pre-intervention and post-intervention sections of table 1 in the results section.
Answer: Thanks for your reminder and we have put the “Characteristics of Study Participants” and “Cardiorespiratory Fitness and Inhibitory Control at Baseline and Post-Test (pre-intervention and post-intervention section)” into two tables.
[3] discussion: please correct this sentence "In the previous study, the increased cortical thickness in LOC.L and PCUN.L is not fully in line with a previous study investigating the effect of long-term endurance training in younger adults as they noticed an increase in cortical thickness in frontal brain regions [51]".
Answer: Sorry, it should be “in the present study”, but not “in the previous study”. And we have revised it as followed.
"In the present study, the increased cortical thickness in LOC.L and PCUN.L is not fully in line with a previous study indicating that long-term endurance training in younger adults resulted in increased cortical thickness in frontal brain regions [51]".
